# Robotic Odor Source Localization via Vision and Olfaction Fusion Navigation Algorithm

**DOI:** 10.3390/s24072309

**Published:** 2024-04-05

**Authors:** Sunzid Hassan, Lingxiao Wang, Khan Raqib Mahmud

**Affiliations:** 1Department of Computer Science, Louisiana Tech University, 201 Mayfield Ave., Ruston, LA 71272, USA; sha040@latech.edu (S.H.); krm070@email.latech.edu (K.R.M.); 2Department of Electrical Engineering, Louisiana Tech University, 201 Mayfield Ave., Ruston, LA 71272, USA

**Keywords:** odor source localization, moth-inspired algorithm, computer vision-based navigation, robot operating system, multi-modal robotics

## Abstract

Robotic odor source localization (OSL) is a technology that enables mobile robots or autonomous vehicles to find an odor source in unknown environments. An effective navigation algorithm that guides the robot to approach the odor source is the key to successfully locating the odor source. While traditional OSL approaches primarily utilize an olfaction-only strategy, guiding robots to find the odor source by tracing emitted odor plumes, our work introduces a fusion navigation algorithm that combines both vision and olfaction-based techniques. This hybrid approach addresses challenges such as turbulent airflow, which disrupts olfaction sensing, and physical obstacles inside the search area, which may impede vision detection. In this work, we propose a hierarchical control mechanism that dynamically shifts the robot’s search behavior among four strategies: crosswind maneuver, Obstacle-Avoid Navigation, Vision-Based Navigation, and Olfaction-Based Navigation. Our methodology includes a custom-trained deep-learning model for visual target detection and a moth-inspired algorithm for Olfaction-Based Navigation. To assess the effectiveness of our approach, we implemented the proposed algorithm on a mobile robot in a search environment with obstacles. Experimental results demonstrate that our Vision and Olfaction Fusion algorithm significantly outperforms vision-only and olfaction-only methods, reducing average search time by 54% and 30%, respectively.

## 1. Introduction

Sensory systems like olfaction, vision, audition, etc., allow animals to interact with the external environment. Among these, olfaction is the oldest sensory system to evolve in organisms [1]. Olfaction allows organisms with receptors for the odorant to identify food, potential mating partners, dangers, and enemies [2]. In some nocturnal mammals like mice, as much as five percent of the genome is devoted to olfaction [3]. Similar to animals, a mobile robot integrated with a chemical sensor can detect odors in the external environment. Robotic odor source localization (OSL) is the technology that allows robots to utilize olfaction sensory inputs to navigate toward an unknown target odor source in the given environment [4]. It has important applications, including monitoring wildfires [5], locating air pollution [6], locating chemical gas leaks [7], locating unexploded mines and bombs [8], locating underground gas leaks [9], marine surveys such as finding hydrothermal vents [10], etc.

Locating an unknown odor source requires an effective OSL algorithm to guide the robot based on sensor observations. Current OSL algorithms include bio-inspired methods that imitate animal olfactory behaviors, engineering-based methods that rely on mathematical models to estimate potential odor source locations and machine learning-based methods that use a trained model to guide the robot toward the odor source. The typical bio-inspired methods include the moth-inspired algorithm that imitates male moths’ mate-seeking behaviors [11], where a robotic agent will follow a ‘surge/casting’ model [12] to reach the odor source. Typical engineering-based methods includes the Particle Filter algorithm [13], where the robot will use historic olfaction reading to predict the odor source location. Finally, typical machine learning-based OSL methods include deep supervised [14] and reinforcement learning-based methods [15].

All of these approaches rely on olfaction (e.g., chemical and airflow) sensing to detect and navigate to the given odor source. However, approaches that rely solely on olfaction sensing struggle in turbulent airflow environments. Simple organisms without vision (e.g., nematodes) use vision-free OSL [16]. But, in addition to humans, vision is a primary sensory modality for odor source localization in mammals, raptors [17], and even invertebrates like fruit flies [18], mosquitoes [19], beetles [20], etc.

To find an unknown odor source location, the sense of smell, i.e., olfaction, is the primary modality on which animals rely [21,22]. Wind sensing is included in the animal olfactory behaviors, such as the mate-seeking behaviors of male moths [23], where a male moth flies against the wind direction to approach the odor source location when it senses high odor concentrations. Somatosensory sensing (i.e., tactile sensing) and thermal sensing are not widely used in the task of finding odor source location. Vision, on the other hand, is important in the task of finding the odor source location [24]. Usually, vision works with olfaction for animals to pinpoint the exact odor source location. In the event of sensing high odor concentration, animals will first use vision to search for possible odor source targets, and, if such targets are not directly visible, animals will rely on olfaction to trace odor plumes and approach the odor source. The final odor source confirmation usually relies on vision. Thus, both vision and olfaction are vital modalities in the task of odor source localization.

Similarly, a robot with both olfaction- and vision-sensing capabilities (e.g., with a camera and chemical sensor) can find an unknown odor source more efficiently, compared to olfaction-only OSL navigation methods. Thus, this project departs from the existing OSL navigation methods in utilizing both robotic vision and olfaction in searching for the odor source location. The core of this project involves designing an algorithm that utilizes both vision and olfaction sensing for locating an unknown odor source location.

The project proposes an effective sensor fusion approach that utilizes a vision method and bio-mimicking olfaction method to guide the robot toward an unknown odor source in a real-world obstacle-ridden search area with both laminar and turbulent airflow setups. Figure 1 shows the proposed method, where we show the developed robot platform equipped with vision and olfaction sensors. The vision sensors include a camera, and the olfaction sensors include a chemical detector and anemometer. It also includes a Laser Distance Sensor (LDS) for obstacle detection. The sensor observations are transmitted to a decision-making model, which is implemented in a remote computer. The model selects Obstacle-Avoid Navigation, Vision-Based Navigation, or Olfaction-Based Navigation behavior based on the sensor readings. In the proposed decision-making model, the robotic vision is achieved by a deep-learning vision model, and the robotic olfaction model is based on a bio-mimicking moth-inspired algorithm. Based on the current sensor reading, the active search behavior calculates the robot heading commands, guiding the robot to approach the odor source location. Finally, the robot executes the heading command, collects new sensor readings at the new location, and repeats the loop until the odor source is detected.

In order to test the performance of our proposed Vision and Olfaction Fusion Navigation algorithm, we conducted 30 real-world OSL experiments using the Olfaction-Only Navigation algorithm, Vision-Only Navigation algorithm, and the proposed Vision and Olfaction Fusion Navigation algorithms in both laminar and turbulent airflow environments. Contributions of this work can be summarized as follows:Introduce vision as an additional sensing modality for odor source localization. For vision sensing, we trained a deep-learning-based computer vision model to detect odor sources from emitted visible plumes;Develop a multimodal Vision and Olfaction Fusion Navigation algorithm with Obstacle-Avoid Navigation capabilities for OSL tasks;Compare the search performance of Olfaction-Only and Vision-Only Navigation algorithms with the proposed Vision and Olfaction Fusion Navigation algorithm in a real-world search environment with obstacles and turbulent airflow setups.

In the remainder of this paper, Section 2 reviews the recent progress of Olfactory-Based Navigation algorithms; Section 3 reviews technical details of the proposed OSL algorithm; Section 4 presents details of the performed real-world experiments; Section 5 includes a discussion on the future research directions based on this work; and, finally, Section 6 includes overall conclusions of the work.

## 2. Related Works

Research on robotic odor source localization (OSL) has gained significant attention in recent decades [25]. Technological advancements in robotics and autonomous systems have made it possible to deploy mobile robots for locating odor or chemical sources. Designing algorithms that mimic the navigation method of biological organisms is a typical approach in robotic odor source localization research. Organisms spanning various sizes rely on scent for locating objects. Whether considering a bacterium navigating an amino acid gradient or a wolf tracking down prey, the ability to follow odors can be crucial for survival.

Chemotaxis is the simplest odor source localization approach in biological organisms, where they rely only on olfaction for navigation. For example, bacteria exhibit chemotaxis by adjusting their movement in response to changes in chemical concentration. When they encounter higher levels of an appealing chemical, their likelihood of making temporary turns decreases, promoting straighter movement. Conversely, in the absence of a gradient or when moving away from higher concentrations, the default turning probability is maintained [26]. This simple algorithm enables single-celled organisms to navigate a gradient of attractive chemicals through a guided random walk. Nematodes [16] and crustaceans [27] also follow chemotaxis-based odor source localization. Early attempts at robotic OSL focused on employing such simple gradient-following chemotaxis algorithms. These methods utilized a pair of chemical sensors on plume-tracing robots, directing them to steer towards higher concentration measurements [28]. Several early studies [29,30,31,32] validated the effectiveness of chemotaxis in laminar flow environments, characterized by low Reynolds numbers. However, in turbulent flow environments with high Reynolds numbers, alternative methods were proposed, drawing inspiration from both complex biological and engineering principles.

Odor-gated anemotaxis navigation is a more complex odor source localization method that utilizes senses of both odor and airflow for navigation. Moths [21,23,33], birds [34,35], etc. are organisms that follow this type of navigation. In particular, mimicking the mate-seeking behavior of male moths led to the development of the moth-inspired method in robotic odor source localization. This method was successfully applied in various robotic OSL scenarios [36]. Additionally, diverse bio-inspired search strategies like zigzag, spiral, fuzzy-inference, and multi-phase exploratory approaches have been introduced [37] in odor-gate anemotaxis-based solutions. Recent bio-inspired OSL navigation methods also aimed to make the search environment more complicated. For instance, chemical plume tracking can be conducted in three-dimensional environments using three-dimensional moth-inspired OSL search [38,39].

Engineering-based methods take a different approach than bio-mimicking algorithms, relying on mathematical models for estimating odor source locations. These methods are often times known as infotaxis [40]. These methods involve constructing source probability maps, dividing the search area into regions, and assigning probabilities indicating the likelihood of containing the odor source. Algorithms for constructing such maps include Bayesian inference, particle filters, stochastic mapping [41], source term estimation [42], information-based search [43], partially observable Markov decision processes [44], combination of infotaxis and Dijkstra algorithm [45], etc. Subsequently, robots are guided towards the estimated source via path-planning algorithms such as artificial potential fields, A-star [46,47]. These models also rely on olfaction sensing for estimating the odor source.

Deep-learning (DL)-based methods are increasingly utilized for OSL experiments. Recent developments involve the use of Deep Neural Networks (DNNs) to predict gas leak locations from stationary sensor networks or employing reinforcement learning for plume-tracing strategies. For instance, Kim et al. [14] trained an RNN to predict potential odor source locations using data from stationary sensor networks obtained through simulation. Hu et al. [15] presented a plume-tracing algorithm based on model-free reinforcement learning, utilizing the deterministic policy gradient to train an actor–critic network for Autonomous Underwater Vehicle (AUV) navigation. Wang et al. [48] trained an adaptive neuro-fuzzy inference system (ANFIS) to solve the OSL problem in simulations, yet real-world validations are necessary to confirm its efficacy. In summary, despite the promising potential of DL technologies, their application in solving OSL problems is still in its early stages and warrants further research. Most DL-based methods are validated in virtual environments through simulated flow fields and plume distributions, necessitating real-world implementations to validate their effectiveness.

Fusing vision with olfaction for odor source localization tasks is common in complex organisms like mice [49,50]. Humans also use vision as a primary sensor for odor source navigation tasks. However, very few works have utilized vision for OSL tasks. Recent advances in computer vision techniques can allow robots to use vision as an important sensing capability for detecting visible odor sources or plumes. The added advantage of vision is that it can allow robots to navigate to odor sources without being affected by sparse odor plumes or turbulent airflow in the navigation path. The main contribution of this paper is designing a dynamic Vision and Olfaction Fusion Navigation algorithm for odor source localization in an obstacle-ridden turbulent airflow environment.

## 3. Materials and Methods

### 3.1. Overview of the Proposed OSL Algorithm

Figure 2 shows the flow diagram of the proposed navigation algorithm. In this work, the initial robot search behavior is the ‘Crosswind maneuver’ behavior, where the robot moves crosswind to detect initial odor plumes. If the robot encounters obstacles in its surroundings, it switches to the ‘Obstacle-Avoid Navigation’ behavior, where the robot moves around to avoid obstacles. During the robot maneuver, the robot seeks valid visual and olfactory detection. If the robot obtains a valid visual detection, it employs Vision-Based Navigation to approach the odor source location. Similarly, if the robot obtains sufficient olfactory detection, it employs the Olfaction-Based Navigation algorithm. If the robot is in the vicinity of the odor source, it is considered as the source declaration, i.e., the end of the search. Otherwise, the robot returns to the default ‘Crosswind maneuver’ behavior and repeats the above process.

In the following section, we present the design of the aforementioned search behaviors, including Crosswind maneuver (Section 3.2), Obstacle-Avoid Navigation (Section 3.3), Vision-Based Navigation (Section 3.4), and Olfactory-Based Navigation (Section 3.5).

### 3.2. Crosswind Maneuver Behavior

In an OSL task, the robot does not have any prior information on the odor source location. Thus, we define a ‘Crosswind maneuver’ behavior as the default behavior, directing the robot to find initial odor plume detection or re-detect odor plumes when valid vision and olfaction observations are absent. Crosswind movement, where the robot heading is perpendicular to the wind direction, increases the chance of the robot detecting odor plumes [51].

In Figure 3, *x*-*o*-*y* is the inertial frame, which represents the fixed global frame. And $x_{b}$-$o_{b}$-$y_{b}$ is the body frame, which is the local frame fixed on the robot. Denote that the wind direction in the inertial frame is $\phi_{Inertial}$; thus, following the ‘Crosswind maneuver’ behavior, the robot target heading in the inertial frame ψ can be defined as
(1)$$\psi_{c}=\phi_{Inertial}+90.$$

In addition, it is worth mentioning that we set the robot’s linear speed as a constant and only changed the target heading commands in the ‘Crosswind maneuver’ behavior to simplify the robot control problem and save search time. The robot then changes the angular velocity to match the target heading.

### 3.3. Obstacle-Avoid Navigation Behavior

The ‘Obstacle-Avoid Navigation’ behavior is activated when the robot moves close to an obstacle object within the search environment, which directs the robot to move around and avoid the obstacles. In this work, the robot employs a Laser Distance Sensor (LDS) to measure the distances from the robot to any obstacles in five surrounding angles as presented in Figure 4. Specifically, we denote laser[x] as the measured distance at angle *x*, including Front (laser[0]), Slightly Left (laser[45]), Slightly Right (laser[315]), Left (laser[90]), and Right (laser[270]). If the obstacle distance in any of the five angles is less than the threshold, the proposed ‘Obstacle-Avoid Navigation’ behavior is activated.

Algorithm 1 shows the pseudo-code for the ‘Obstacle-Avoid Navigation’ behavior. The main idea is to identify the relative location of obstacles to the robot and command the robot to move around to avoid obstacles. Specifically, the robot initially sets the linear velocity and angular velocity as vc and ωc, respectively. Positive values in vc and ωc represent forward and left rotation, respectively, and negative values represent backward and right rotation, respectively. Initial values of vc and ωc are set as 0.6 m/s and 0 rad/s in this work.
**Algorithm 1** ‘Obstacle-Avoid Navigation’ Behavior1: Set robot linear velocity as vc=0.6 m/s2: Set robot angular velocity as ωc=0 rad/s3: **if** 
laser[0]>thr **then**4:     ωc=0 rad/s5: **else**6:      vc=0 m/s and ωc=0 rad/s7:      **if** (laser[45]>thr)∨(laser[315]>thr) **then**8:            **if** laser[45]>laser[315] **then**9:               ωc=0.5 rad/s10:          **else**11:             ωc=−0.5 rad/s12:          **end if**13:     **else if** (laser[90]>thr)∨(laser[270]>thr) **then**14:          **if** laser[90]>laser[270] **then**15:              ωc=0.5 rad/s16:          **else**17:              ωc=−0.5 rad/s18:          **end if**19:     **else**20:           vc=−0.5 m/s21:     **end if**22: **end if**


In the ‘Obstacle-Avoid Navigation’ behavior, the robot will always check if there is a clear path in the Front direction, i.e., laser[0]>thr (thr is the threshold for obstacle detection, 0.75 m in this work), and, if it is true, the robot will move forward with $\omega_{c}=0$ rad/s. If the Front is blocked, the robot will stop and check Slightly Left or Slightly Right for a clear path ((laser[45]>thr)∨(laser[315]>thr)). If there is a clear path in either of these two directions, the robot will compare clearance in Slightly Left and Slightly Right and rotate left (i.e., $\omega_{c}=0.5$ rad/s) or right (i.e., $\omega_{c}=-0.5$ rad/s) to face the greater clearance. If there is no clearance in Slight Left or Slight Right, the robot will check Left and Right for a clear path ((laser[90]>thr)∨(laser[270]>thr)). If there is a clear path, the robot will compare Left and Right clearance (laser[90]>laser[270]) and rotate left ($\omega_{c}=0.5$ rad/s) or right ($\omega_{c}=-0.5$ rad/s) to face the greater clearance. If there is no clear path in all five directions, the robot will move back ($v_{c}=-0.5$ m/s) to escape the dead end.

It should be mentioned that the proposed navigation algorithm does not have access to the global map of the search environment, the locations of obstacles, and destination odor source. Therefore, planning-based obstacle avoidance algorithms like Artificial Potential Field algorithm [52], A* algorithm [53], Dijkstra algorithm [54], etc. are not applicable in our problem. In such partially observable environments, using discrete behavior control would be the best option. A similar approach was employed in [55]. Compared to classic motion-planning algorithms (e.g., DJ, A-star, APF), our proposed obstacle avoidance algorithm has following advantages:It does not rely on prior knowledge of the global map nor on location of the obstacles or the destination;Compared to most deep-learning-based navigation planners, it requires less inference time.

### 3.4. Vision-Based Navigation

In this work, we employ vision as the main approach to detect odor sources within the search environment. Vision sensing allows the robot to detect the plume source location in its visual field and approach it directly. Olfaction-Only Navigation methods often rely on airflow direction for navigating to the odor source. This can lead to failure in turbulent airflow environments. Given that visual sensing is not guided by airflow direction, combining it with Olfaction-Based Navigation can allow the robot to find the odor source in turbulent airflow environments.

The proposed Vision-Based Navigation relies on computer vision techniques. Specifically, we train a deep-learning-based object detection model, i.e., YOLOv7, to detect vapors emitted from the odor source. Vapors can be considered as a common and distinct feature for the odor source object, such as smoke for fire sources, chemical plumes for chemical leaks or hydrothermal vents, etc. It should be mentioned that, if the odor source does not have a distinct plume feature (i.e., transparent vapors), the robot can still find the odor source using the Olfaction-Based Navigation behavior of the proposed Vision and Olfaction Fusion algorithm. Additionally, we have provided real-world performance comparison between the Olfaction-Based Navigation, Vision-Based Navigation and the proposed navigation algorithms.

In the proposed vision-sensing method, we trained a YOLOv7 model to detect odor plumes in the continuously captured images. To generate training images, we extracted 243 observation frames with a resolution of 640×480 while the Turtlebot was approaching the odor plumes at a variety of angles and in different lighting conditions. Figure 5 shows two sample frames used for training the vision model. These data were split into training, validation, and testing datasets for training the model. Roboflow [56] was utilized as the annotation tool for accurate bounding boxes and polygon delineation.

To assess YOLOv7 performance, diverse predefined augmentation techniques in Roboflow were systematically applied to ‘Dataset-1’. These included rotation (−10° to +10°), shear (±15° horizontally and vertically), hue adjustment (−25° to +25°), saturation adjustment (−25% to +25%), brightness adjustment (−25% to +25%), exposure adjustment (−25% to +25%), blur (up to 2.5 px), and noise (up to 1% of pixels). Post-augmentation, the resulting augmented dataset, labeled as ‘Dataset-3’, enriched the training set for a comprehensive evaluation of YOLOv7’s robustness in detecting prescribed odor plumes. We set the number of training epochs to 100, with a batch size of 16. The resulting training accuracy was 98% and testing accuracy was 93%.

The implemented vision model returns a box bounding the plume in the image if it detects an odor plume. The output of the model also includes the horizontal and vertical location of the plume bounding box. If the model returns a plume bounding box, the robot continues moving forward (i.e., $v_{c}=0.5$ m/s) and checks if the horizontal location of the bounding box is in the left or the right half of the image. The model requires less than 1 s to generate output in our remote computer. The robot sends 30 image frames per second, and the robot picks every 30th frame as the input to the vision model.

Equation (Equation 2) calculates the robot’s heading
(2)$$\omega_{c}=\left\{\begin{array}{cc} 1 & 0.5\;m/s\;\mathrm{if}\;c<\frac{w}{2}\\ 2 & -0.5\;m/s\;\mathrm{if}\;c>\frac{w}{2}, \end{array}\right.$$
where *c* is the horizontal mid-point of the bounding box and w is the horizontal resolution of the captured image. If the bounding box is in the left half of the image (i.e., $c<\frac{w}{2}$), the robot rotates left (i.e., $\omega_{c}=0.5$ rad/s) to face the plume. Otherwise, it rotates right ($\omega_{c}=-0.5$ rad/s) to face the plume.

### 3.5. Olfaction-Based Navigation

If there is no valid visual detection but the robot can sense above-threshold odor concentration, Olfaction-Based Navigation is employed to guide the robot to approach the odor source location.

Specifically, the proposed Olfaction-Based Navigation algorithm commands the robot to move upwind to approach the odor source location. This behavior is analogous to the ’Surge’ behavior of the bio-mimicking moth-inspired navigation OSL algorithm [57]. In this behavior, the robot’s linear velocity is fixed at $v_{c}=0.6$ m/s and the heading command, i.e., $\psi_{c}$, is calculated as
(3)$$\psi_{c}=\phi_{Inertial}+180.$$

The robot will switch back to Vision-Based Navigation once there is a valid vision detection.

## 4. Experiment Results

### 4.1. Search Area

Figure 6a shows the two-dimensional 8.2 m × 3.3 m search area, and the Figure 6b shows the robot platform. Two obstacles were placed in the search area to simulate a complex real-world search environment. Ethanol vapor was used as the odor source as it is not toxic. Ethanol is also commonly implemented in OSL research [58]. A humidifier dispersed ethanol vapor constantly as the odor plume. In our work, the fans were placed to create both laminar and turbulent airflow environments in the search area. In laminar flow environments, only one fan was employed, and it was placed behind the odor source to accelerate the plume diffusion rate and create a unified wind direction field (as presented in Figure 7a). In turbulent flow environments, we used two fans and placed them at perpendicular positions to create a mixed and turbulent wind field (as presented in Figure 7b). In turbulent flow environments, the wind direction is not unified but mixed and turbulent. In such environments, correctly finding the odor source relying only on olfaction is harder.

### 4.2. Mobile Robot Configuration

Turtlebot3 mobile robot platform was used in this work. Its built-in sensors include Raspberry Pi Camera, a 360-degree LiDAR sensor for sensing, and a DYNAMIXEL diver for navigation. The onboard OpenCR controller allows the Turtlebot3 to be paired with additional sensors for increasing its functionalities.

Table 1 shows the built-in and added sensors for OSL experiments. Raspberry Pi Camera V2 was used for image capture, LDS-02 Laser Distance Sensor was used for obstacle detection, WindSonic Anemometer was used for wind speed and wind direction measurements in the body frame, and MQ3 alcohol detector was used for detecting chemical plume concentration.

Turtlebot3 has Raspberry Pi 4 as the CPU, which has limited computing power. It utilizes Ubuntu and Robot Operating System (ROS). Ubuntu allows connection capabilities with a remote computer. ROS allows custom programs in the remote computer to subscribe to specific sensor readings from the robot and publish heading commands back to the robot in real time. ROS supports both Python and C++ programming languages. Figure 8 presents the proposed system configuration for the robotic system, which includes a robotic agent, i.e., Turtlebot3, onboard controller, and a ground station, i.e., a remote Personal Computer (PC). For this study, Ubuntu 20.04 and ROS Noetic were installed in both the robot and the paired remote computer for controlling the robot. A local area network was used to connect the robot to the remote PC.

### 4.3. Experiment Design

To determine the effectiveness of the proposed Vision and Olfaction Fusion Navigation algorithm, we tested the performance of Olfaction-Only Navigation and Vision-Only Navigation algorithms. Figure 9 shows the flow diagram of the two navigation algorithms. In the Olfaction-Only Navigation algorithm, the robot used the Crosswind maneuver behavior (Section 3.2), Obstacle-Avoid Navigation behavior (Section 3.3), and Olfaction-Based Navigation behavior (Section 3.5). In the absence of sufficient chemical concentration, the robot followed Crosswind maneuver behavior to maximize the chance of detecting sufficient plume concentration. If there were obstacles in the robot’s path, it followed Obstacle-Avoid Navigation behavior to circumvent the obstacles. If sufficient odor concentration was detected and there were no obstacles in the robot’s path, it followed Olfaction-Based Navigation behavior to reach the odor source.

In the Vision-Only Navigation algorithm, the robot used the Crosswind maneuver behavior (Section 3.2), Obstacle-Avoid Navigation behavior (Section 3.3), and Vision-Based Navigation behavior (Section 3.4). In the absence of valid plume vision, the robot followed Crosswind maneuver behavior to maximize the chance of detecting valid plume vision. If there were obstacles in the robot’s path, it followed Obstacle-Avoid Navigation behavior to circumvent the obstacles. If the robot detected a valid plume visual and there were no obstacles in the robot’s path, it followed Vision-Based Navigation behavior to reach the odor source.

These three algorithms were tested in two airflow environments, including the e1—laminar airflow environment that used one electric fan—and the e2—turbulent airflow environment that used two perpendicularly placed electric fans. Thus, a total of six experimental setups were designed, i.e., three navigation methods in two airflow environments, to test the effectiveness of the proposed fusion model. Five experimental runs were conducted for each of the six experimental setups, totaling 30 trial runs. We used the same five starting positions to initialize the test runs. Figure 7 shows the five starting positions and the two airflow setups for the experimental runs.

### 4.4. Source Declaration

The robot is considered successful if the robot position is within 0.9 m of the odor source location. But, if the robot fails to reach the odor source within 200 s, the trial run is considered as a failure.

### 4.5. Sample Trials

Figure 10 shows the robot trajectory and snapshots of the Vision and Olfaction Fusion Navigation trial run in a turbulent airflow environment. In this run, the robot initialized at *t* = 1 s, found sufficient chemical concentration, and started following Olfaction-Based Navigation. At *t* = 22 s, the robot detected valid visual detection of the odor plumes and started to follow Vision-Based Navigation. At *t* = 49 s, the robot faced the second obstacle and started to follow Obstacle-Avoid Navigation behavior. It avoided the obstacle, re-detected plume vision, and started to follow Vision-Based Navigation until it reached the odor source at *t* = 72 s.

### 4.6. Repeated Experimental Trials

Table 2 shows the run times of the 30 trial runs, i.e., 5 trial runs using 1 of 3 navigation algorithms in 2 airflow environments. Figure 11 shows the combined robot trajectories of the three navigation algorithms in the two airflow environments. Table 3 summarizes the repeated test results in terms of success rate, average search time, and average traveled distance. We can observe from the results that the proposed Vision and Olfaction Fusion Navigation algorithm has the highest success rate, the lowest average search time, and the lowest average distance traveled among the three methods. This is critical in real-world odor source localization applications, as we want the robot to find odor sources as quickly as possible.

The Olfaction-Only Navigation algorithm uses airflow direction to navigate toward the odor source. It performed well in laminar airflow environments—the robot followed relatively direct airflow towards the odor source. However, in turbulent airflow environments, the robot was diverted by the complex airflow directions and often failed to reach the odor source by the designated time limit. Vision-Based Navigation performed poorly in both laminar and turbulent airflow environments. Because of the obstacle placement, the robot had no vision of the plume from the starting position. It needed to follow the Crosswind maneuver and Obstacle-Avoid Navigation behaviors until it had valid plume vision. In most runs, the robot’s 200 s time limit was over before it could find and navigate to the odor source.

Vision and Olfaction Fusion Navigation algorithm test runs were consistently successful in both laminar and turbulent airflow environments. The Crosswind maneuver and Olfaction-Based Navigation led the robot toward the odor source, which allowed the robot to detect plume vision. Once it started to follow Vision-Based Navigation, the robot was not affected by turbulent airflow. Using dual-modality sensing could also reduce the false negative cases. In the proposed Vision and Olfaction Fusion Navigation algorithm, the two sensory modalities work together to determine the final odor source location. The combination increases the probability of reducing overall false detection cases, i.e., if one modality provides false detection, the probability is high that another modality can correct it.

## 5. Discussion and Future Research Directions

The proposed Vision and Olfaction Fusion algorithm can improve odor source localization performance in diverse environments. We set up the presented experimental field to mimic indoor environments with obstacles and odor sources. Therefore, the experiment results can be generalized to other real-world indoor odor source localization scenarios, such as detecting indoor gas leaks in office or household environments with obstacles and potential gas sources. It is also possible to extend the proposed method to outdoor applications, such as detecting wildfire locations using both vision (flame detection) and olfaction (smoke or other fire-related gases).

The result of our experiment indicates that vision sensing is a promising addition to olfaction sensing in robotic odor source localization research. We summarize the following significances of the proposed work:Integration of vision and olfaction in odor source localization tasks: Our proposed navigation algorithm integrates both vision and olfaction in odor source localization tasks. Compared to traditional Olfaction-Only Navigation algorithms, including bio-inspired methods [12], engineering-based methods [13,45], and machine-learning-based methods [14,15], the addition of vision advances the boundaries of current OSL navigation algorithms;Odor source localization in complex environments with obstacles: While most traditional olfactory-based navigation algorithms do not consider obstacles in the search environments (e.g., [12]), our proposed method can guide the robot to find the odor source in complex environments with obstacles. Thanks to the proposed hierarchical control algorithm, the robot can dynamically coordinate among Vision-Based Navigation, Olfaction-Based Navigation, and obstacle avoidance behaviors;Real-world experiments and results: Many prior works (e.g., [15]) only validated their algorithms in simulation environment without validating them in real-world environments. However, simulation environments cannot always represent real-world scenarios due to the gap between the simulation and real-world environments. In this work, we implemented the proposed odor source localization algorithm in real-world settings, showed it in real-world settings, and validated its effectiveness in real-world environments with obstacles and turbulent airflow.

A number of improvements can be made to the proposed OSL algorithm in the future. Firstly, the proposed navigation algorithm follows a homogeneous Crosswind maneuver behavior for finding odor plumes. The search behavior does not take into account past vision or olfaction sensing history. Similarly, the moth-inspired algorithm used in this paper only uses current olfaction readings for finding the odor source, whereas engineering-based solutions like the Particle Filter utilize past sensor readings for estimating the odor source and plume location. Thus, future research scope includes pairing engineering-based Olfaction-Based Navigation with Vision-Based Navigation for improved Crosswind maneuver and Olfaction-Based Navigation. The implemented Obstacle-Avoid Navigation algorithm in this paper also relies only on the current laser readings to sense and circumvent obstacles. In this case, reactive path-planning algorithms, which include Fuzzy Logic, Neural Networks, bug algorithms, etc. [59], can be adopted for more efficient Obstacle-Avoid Navigation behavior. Additionally, the future scope of this robot platform includes using machine-learning algorithms for calculating robot headings. For instance, the reinforcement-learning (RL) [60] and supervised-learning [61] methods can be used for Olfactory-Based Navigation in robots. Transformer-based Vision–Language and Vision–Language–Action (VLA) models are gaining traction as a prevalent approach in robotics. Recent applications of such a model include the PaLM-E model [62] and the RT-2 [63]. Exploring the possibilities of the Vision–Language models as the primary decision-maker for multimodal odor source localization is another exciting possibility in OSL research.

## 6. Conclusions

The combination of computer vision and robotic olfaction provides a more comprehensive observation of the environment, enabling the robot to interact with the environment in more ways and enhancing navigation performance. This paper proposes the incorporation of vision sensing in OSL. Specifically, the paper proposes a Vision and Olfaction Fusion Navigation algorithm with Obstacle-Avoid Navigation capability for 2-D odor source localization tasks for ground mobile robots.

For conducting real-world experiments to test the proposed algorithm, a robot platform based on the Turtlebot3 mobile robot was developed with olfaction- and vision-sensing capabilities. The proposed navigation algorithm had five behaviors, i.e., Crosswind maneuver behavior to find odor plume, Obstacle-Avoid Navigation behavior to circumvent obstacles in the environment, Vision-Based Navigation to approach the odor source using vision sensing, Olfaction-Based Navigation to approach the odor source using olfaction sensing, and source declaration. For the Vision-Based Navigation behavior, a YOLOv7-based vision model was trained to detect visible odor plumes. For Olfaction-Based Navigation behavior, we used moth-inspired algorithm.

To test the performance of the proposed Vision and Olfaction Fusion Navigation algorithm, we tested the performance of the Olfaction-Only Navigation algorithm, Vision-Only Navigation algorithm, and the proposed Vision and Olfaction Fusion Navigation algorithm separately in real-world experiment setups. Furthermore, we tested the performance of the three navigation algorithms in laminar and turbulent airflow environments to compare their strengths. We used five predefined starting robot positions for each navigation algorithm and repeated them for both airflow environments, resulting in 30 total experimental runs.

The search results of the OSL experiments show that the proposed Vision and Olfaction Fusion Navigation algorithm had a higher success rate, lower average search time, and lower average traveled distance for finding the odor source compared to Olfaction-Only and Vision-Only Navigation algorithms in both laminar and turbulent airflow environments.

## Figures and Tables

**Figure 1 sensors-24-02309-f001:**
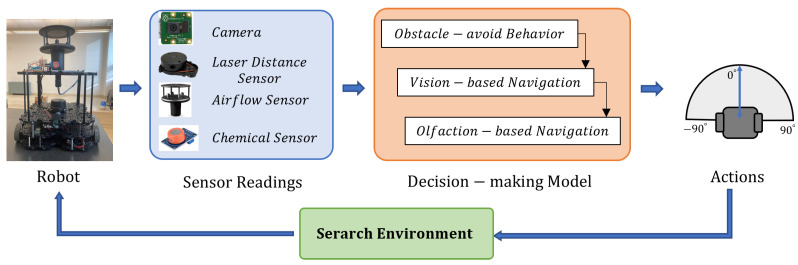
Flow diagram of the proposed method for the OSL experiment. We utilized the Turtlebot3 robot platform. We equipped it with a camera, Laser Distance Sensor, airflow sensor, chemical sensor, etc. The robot utilizes 3 navigation behaviors—Obstacle-Avoid Navigation, Vision-Based Navigation, and Olfaction-Based Navigation to output robot heading and linear velocity.

**Figure 2 sensors-24-02309-f002:**
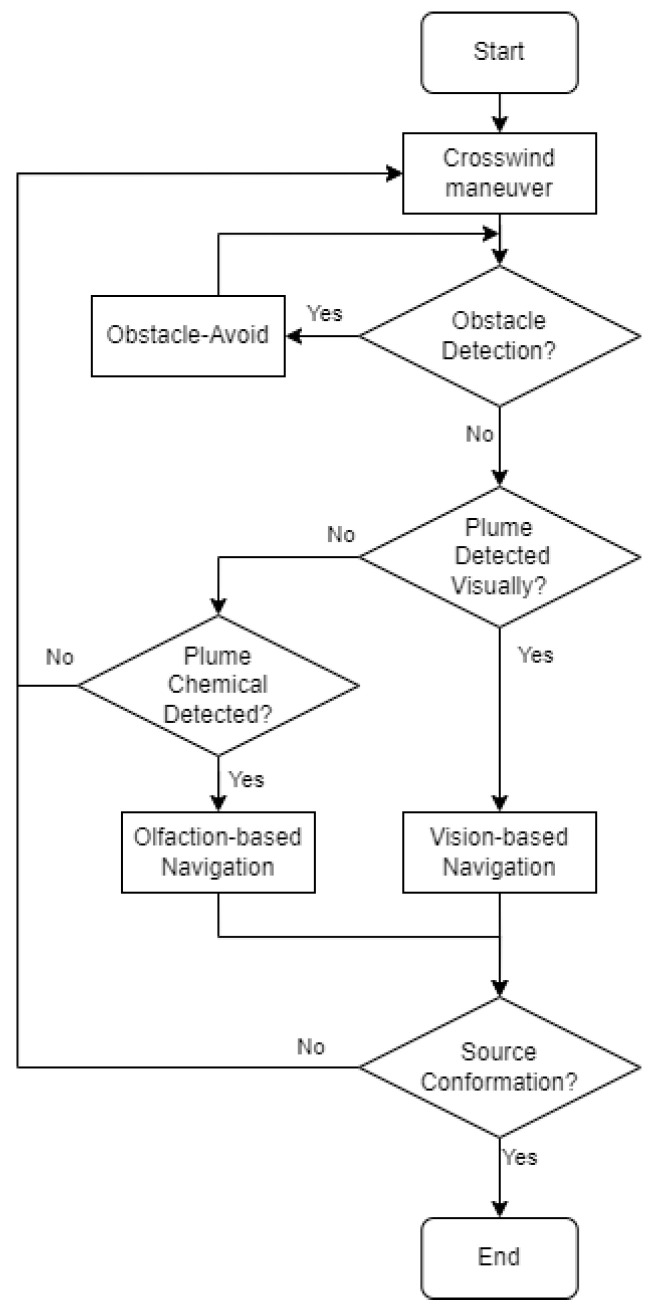
The flow diagram of the proposed OSL algorithm. There are four navigation behaviors, including ‘Crosswind maneuver’, ‘Obstacle-Avoid Navigation’, ‘Vision-Based Navigation’, and ‘Olfaction-Based Navigation’.

**Figure 3 sensors-24-02309-f003:**
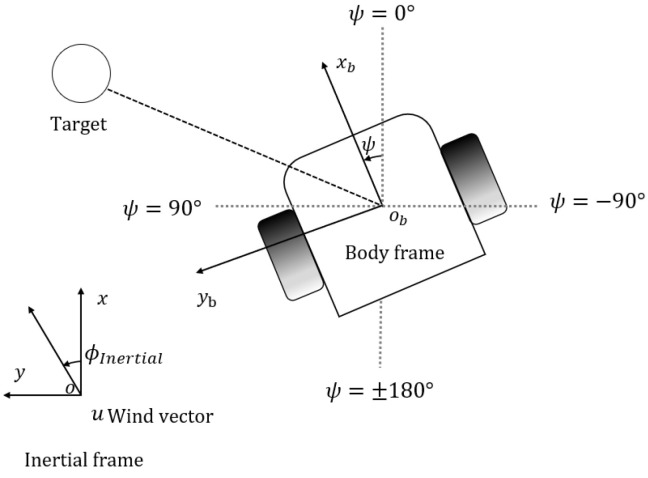
Robot notations. Robot position (x,y) and heading ψ are monitored by the built-in localization system. Wind speed *u* and wind direction are measured from the additional anemometer in the body frame. Wind direction in inertial frame $\phi_{Inertial}$ is derived from robot heading ψ and wind direction in body frame.

**Figure 4 sensors-24-02309-f004:**
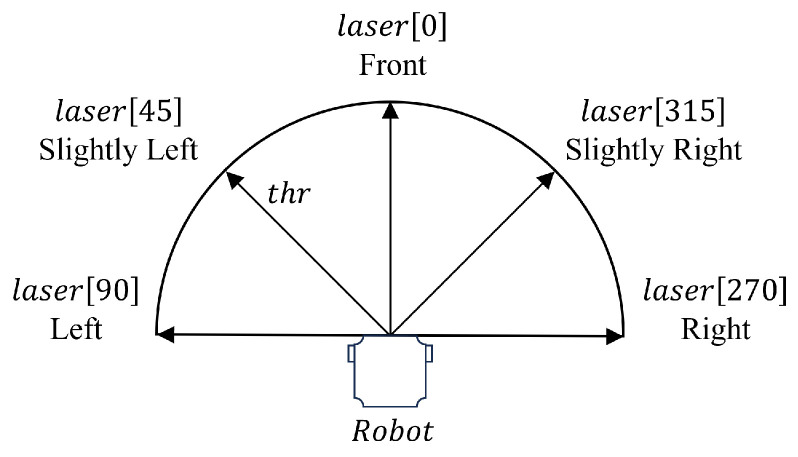
Five directions in the robot’s laser distance sensing, including Left, Slightly Left, Front, Slightly Right, and Right. laser[x] denotes the distance between the robot and the object at the angle *x*, which is measured from the onboard laser distance sensor.

**Figure 5 sensors-24-02309-f005:**
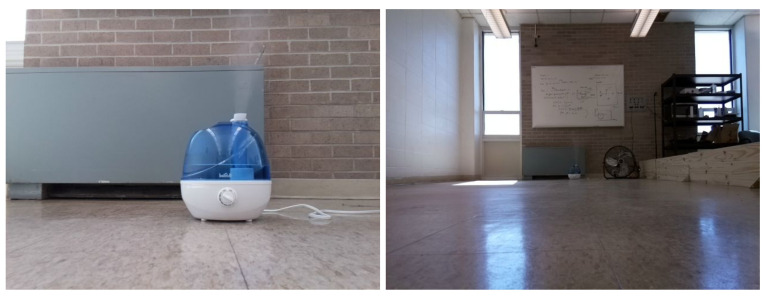
Two sample frames that include humidifier odor plumes in different lighting and spatial conditions. The frames are sampled out of the total 243 frames used for training the vision model. All of the frames were captured by the Turtlebot robot in the experiment area.

**Figure 6 sensors-24-02309-f006:**
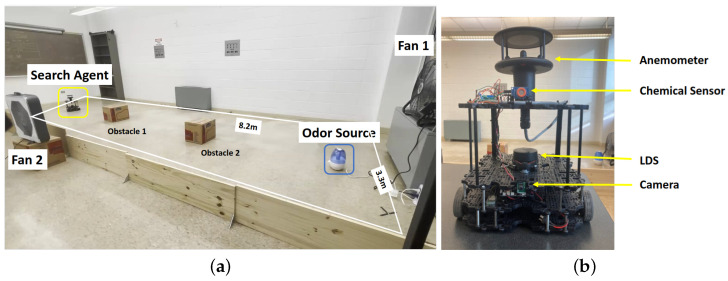
(**a**) The experimental setup. The robot is initially placed in a downwind area with the objective of finding the odor source. A humidifier loaded with ethanol is employed to generate odor plumes. Two electric fans are placed perpendicularly to create artificial wind fields. Two obstacles are placed in the search area. (**b**) The Turtlebot3 waffle pi mobile robot is used in this work. In addition to the camera and Laser Distance Sensor, the robot is equipped with a chemical sensor and an anemometer for measuring chemical concentration, wind speeds, and directions.

**Figure 7 sensors-24-02309-f007:**
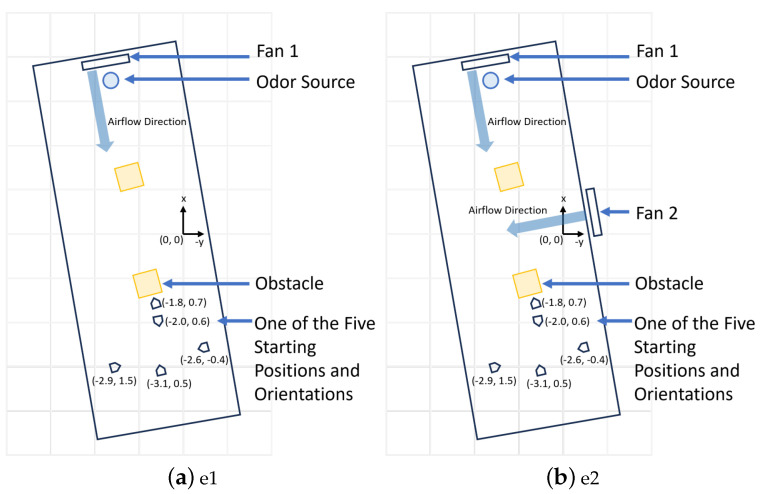
(**a**) The schematic diagram of the search area with e1−laminar airflow setup. The five robot starting positions are used for testing the performance of the Olfaction-Based Navigation, Vision-Based Navigation, and Vision and Olfaction Fusion Navigation tests. (**b**) The schematic diagram of the search area with e2−turbulent airflow setup.

**Figure 8 sensors-24-02309-f008:**
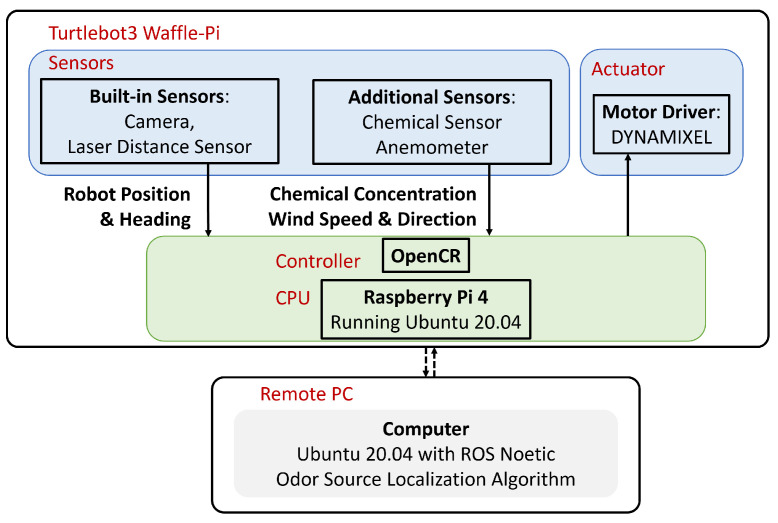
System configuration. This system contains two main components, including the Turtlebot3 and the remote PC. The solid connection line represents physical connection, and the dotted connection line represents wireless link.

**Figure 9 sensors-24-02309-f009:**
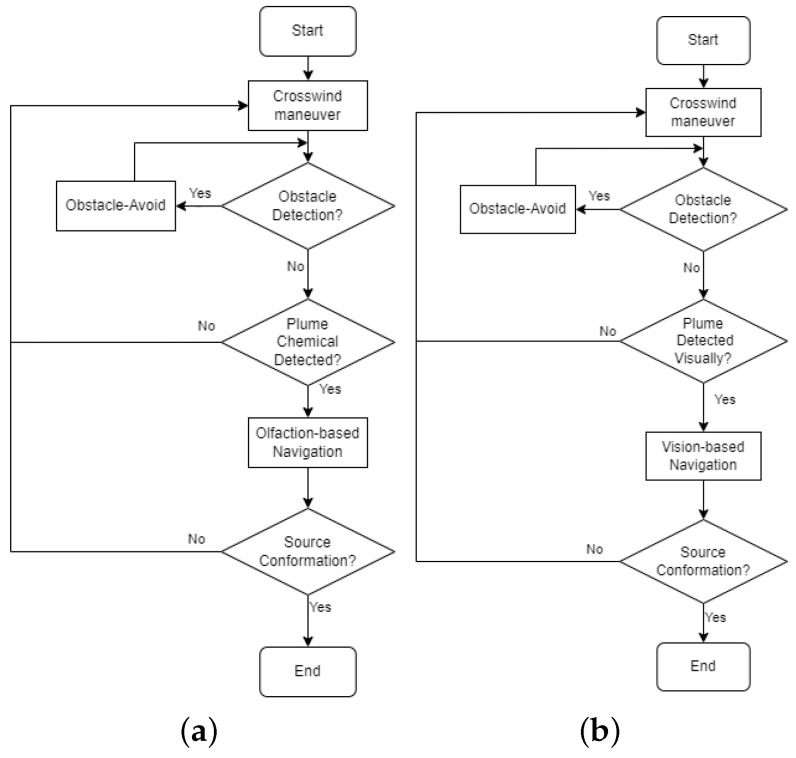
(**a**) The flow diagram of the Olfaction-Only Navigation algorithm. There are three navigation behaviors, including ‘Crosswind maneuver’, ‘Obstacle-Avoid Navigation’, and ‘Olfaction-Based Navigation’. (**b**) The flow diagram of the Vision-Only Navigation algorithm. There are three navigation behaviors, including ‘Crosswind maneuver’, ‘Obstacle-Avoid Navigation’, and ‘Vision-Based Navigation’.

**Figure 10 sensors-24-02309-f010:**
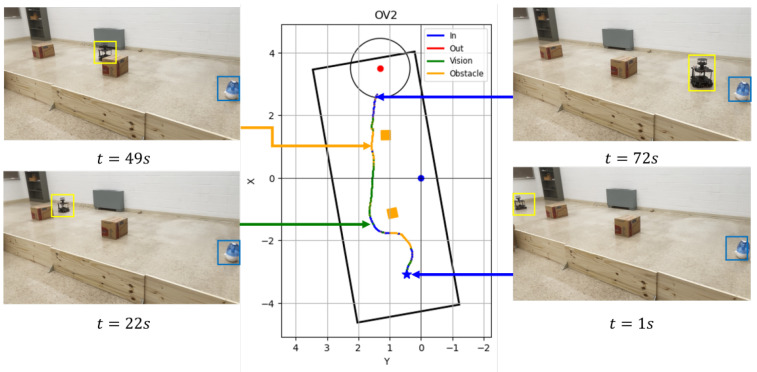
Robot trajectory graphs and snapshots of OSL tests with the Vision and Olfaction Fusion Navigation algorithm in turbulent airflow environment.

**Figure 11 sensors-24-02309-f011:**
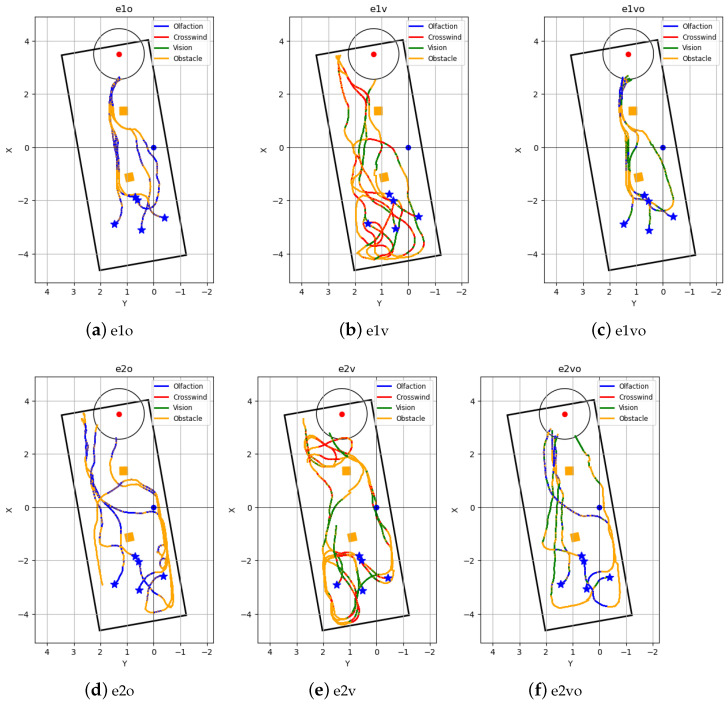
Robot trajectories of repeated tests in six navigation algorithm and airflow environment combinations. Trajectories in laminar airflow environments are (**a**) e1o—Olfaction-Only Navigation algorithm, (**b**) e1v—Vision-Only Navigation algorithm, and (**c**) e1vo—Vision and Olfaction Fusion Navigation algorithm. Trajectories in turbulent airflow environment are (**d**) e2o—Olfaction-Only Navigation algorithm, (**e**) e2v—Vision-Only Navigation algorithm, (**f**) e2vo—Vision and Olfaction Fusion Navigation algorithm. The behaviors that the robot was following under the three navigation algorithms are shown in the trajectory. These behaviors include Crosswind (Crosswind maneuver behavior), Obstacle (Obstacle-Avoid Navigation behavior), Olfaction (Olfaction-Based Navigation behavior), and Vision (Vision-Based Navigation behavior). Five robot starting positions are highlighted with a blue star, the obstacles are the orange boxes, and the odor source is the red point with the surrounding circular source declaration region.

**Table 1 sensors-24-02309-t001:** Type, name, and specification of the built-in camera, laser distance sensor, and added anemometer and chemical sensor.

Source	Sensor Type	Module Name	Specification
Built-in	Camera	Raspberry Pi Camera v2	Video Capture: 1080p30, 720p60 and VGA90.
	Laser Distance Sensor	LDS-02	Detection Range: 360-degree. Distance Range: 160∼8000 mm.
Added	Anemometer	WindSonic, Gill Inc.	Speed: 0–75 m/s. Wind direction: 0–360 degrees.
	Chemical Sensor	MQ3 alcohol detector	Concentration: 25–500 ppm.

**Table 2 sensors-24-02309-t002:** Search time of the Vision-Only, Olfaction-Only, and Proposed Vision and Olfaction Fusion Navigation algorithms. The notation (-) indicates that the search time is beyond the limit, which is 200 s in this work.

	Robot Initial Position (x, y), Orientation (z, w)	Olfaction-Only Navigation Algorithm (s)	Vision-Only Navigation Algorithm (s)	Vision and Olfaction Fusion Navigation Algorithm (s)
Laminar Airflow Env.	(−2.9, 1.5), (−0.6, 1.0)	**63.1**	-	63.9
	(−3.1, 0.5), (0.0, 35.0)	71.3	149.3	**69.9**
	(−2.6, −0.4), (0.7, 0.7)	74.3	-	**67.5**
	(−2.0, 0.6), (1.0, −0.1)	**73.8**	-	75.7
	(−1.8, 0.7), (0.0, 0.1)	**59.1**	-	61.1
Turbulent Airflow Env.	(−2.9, 1.5), (−0.6, 1.0)	-	-	**64.0**
	(−3.1, 0.5), (0.0, 35.0)	-	-	**113.1**
	(−2.6, −0.4), (0.7, 0.7)	196.4	-	**130.7**
	(−2.0, 0.6), (1.0, −0.1)	-	102.8	**131.9**
	(−1.8, 0.7), (0.0, 0.1)	72.3	-	**68.5**

**Table 3 sensors-24-02309-t003:** Result statistics, i.e., success rate and average search time of Vision-Based Navigation, Olfaction-Based Navigation, and the Proposed Vision and Olfaction Fusion Navigation Algorithms.

Navigation Algorithm	Airflow Environment	Success Rate	Avg. Search Time (s)	Avg. Travelled Dist. (m)
Olfaction-Only	Laminar	**5/5**	68.3	**6.1**
	Turbulent	2/5	134.4	9.7
Vision-Only	Laminar	1/5	149.3	11.7
	Turbulent	1/5	102.8	13.7
Vision and Olfaction Fusion	Laminar	**5/5**	**67.6**	6.2
	Turbulent	**5/5**	**101.6**	**7.8**

## Data Availability

The raw data supporting the conclusions of this article will be made available by the authors on request.

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
