# Peer review of "Robotic Odor Source Localization via Vision and Olfaction Fusion Navigation Algorithm"

_sensors, 2024, doi:10.3390/s24072309_

Round 1

Reviewer 1 Report

Comments and Suggestions for Authors

The manuscript considers interesting issues, but the results are limited to the presented experiment. I don't see any generalization of the presented issues. 

Moreover, e.g., Fig. 10 has a lot of data that are unclear and show only variants of the recorded data. You have to modify it.

You should point out at least a few more significant results than the general conclusions given in the final section of the manuscript.

Author Response

Thanks for reviewing our manuscript. We address your questions in the following document. 

Reviewer 2 Report

Comments and Suggestions for Authors

This paper proposed vision and olfaction fusion navigation algorithm. Because odor source localization is an engineeringly difficult task, the reviewer thinks that the approach of experimenting using this method as in this paper is good for this research field. However, there are major problems in designing algorithms and interpreting results. Therefore, the current manuscript format is not acceptable because there are many items that are weakly claimed or insufficiently explained. Specific comments are shown below.

- Page 2, L41-42: The authors should add a citation indicating what studies are available.

- Page 2, L43-57: This paragraph claims that organisms generally use vision and smell, but it is known that animals other than humans have poor eyesight. Hence, animals should use other modalities such as wind, tactile, heat, and so on. The authors should make a convincing argument as to why you should use vision among the multiple candidate modalities. The current paragraph is not fair.

- Page 3, L124-126: Recently, a 3D algorithm in turbulent environments have been demonstrated with a palm-sized drone.

Shigaki, S., Yoshimura, Y., Kurabayashi, D., & Hosoda, K. (2022). Palm-Sized Quadcopter for Three-Dimensional Chemical Plume Tracking. IEEE Transactions on Instrumentation and Measurement, 71, 1-12.

-Page 3-4, L127-136: It has been reported that OSL can be efficiently achieved by switching between the Infotaxis and Dijkstra algorithms in the obstacle domain depending on the situation, and I recommend that you should cite its research.

Luong, D. N., & Kurabayashi, D. (2023). Odor Source Localization in Obstacle Regions Using Switching Planning Algorithms with a Switching Framework. Sensors, 23(3), 1140.

- Page5, Eq. (1): Please explain in more detail. What does the angle of the output represent? Is it the attitude in the robot coordinate system? And what exactly is the angle in the inertial frame?

-Page5, Chap3.3: The big question here is why discrete behavior control even though LiDAR is installed? As you know, with LiDAR, it is easy to create spatial maps from point cloud data, and obstacle avoidance can be done more smoothly. Therefore, it should also be possible to avoid behind obstacles, which would be a disadvantage for olfactory navigation. Despite the system having rich data, it is necessary to justify the meaning of using very classical control.

- Page5, Chap3.4: This time, the authors are using yolo to recognize objects, which is supervised learning. This time, what are you using as an odor source to make the robot learn? Doese the robot learn vapor? Or dose it learn the humidifier in Fig. 4? Considering the practical aspect, it is most likely an unknown odor source, and supervised learning is difficult to use. Moreover, the authors mentioned that the odor source is a vapor-emitting object, as you know, there are also many colorless and transparent chemicals. In addition, in your case, you employed something like a scent diffuser, but it is quite possible that it is in a completely different container and is not emitting vapor, or that the paper is coated with chemicals. Given this, this technique is quite limited and is not expected to work in situations beyond laboratory-level experiments. Based on the current explanation, the reviewer cannot agree with this method at all. The paper should be rewritten to justify the method, and the limitations of the research should be made clear.

- Page8, Chap3.6: This is a condition for the end of the experiment, not an automatic "source declaration" by the robot. Therefore, it is inappropriate to put it here. It should be placed in the design of the experiment.

- Page8, Figure 5: The robot picture is small and difficult to see. It is recommended that the picture of the robot be displayed independently.

- Page13, Figure 10: Fig. 10 and Fig. 11 are exactly the same result, so there is no increase in the amount of information. Fig. 10 (or Fig. 11) should be deleted.

- Page13, Chap4.6: This is not a STATISTICAL ANALYSIS at all. What specific statistical method is used? Is there any significant difference in success rate or search time depending on the method used? I think that 5 trials are too small for statistical analysis.

- Page13, Chap4.5 and 4.6: There was no discussion of why combining vision and olfaction would improve the results, just showing the results. The authors should discuss how the combination contributes to the function of OSL.

- Page14, L361-362: It is unfair to calculate the average search time by assuming that a failed trial is 200 seconds. It is intentionally increasing the search time for conditions with many failures. The search time should be calculated only for successful trials.

Comments on the Quality of English Language

Some sentences are difficult to read in places and need minor corrections.

Author Response

(The authors gave the same response as above.)

Reviewer 3 Report

Comments and Suggestions for Authors

The author proposed an olfaction/vision fusion system for object localization in an open space. Here are my comments:

1. The system proposed could improve false negative cases. Please add some elaboration on how the system remedies the false positive cases. 

2. How the setup positions of fan will affect the experiment results?

Comments on the Quality of English Language

The overall presentation of the manuscript is satisfactory

Author Response

(The authors gave the same response as above.)

Round 2

Reviewer 2 Report

Comments and Suggestions for Authors

Most revisions have been made to the comments.
However, the way the results in Table 3 are presented may confuse the reader. I think that "Combined" is the sum of Laminar and Turbulent results, is that right? If so it is not "Airflow Environment". In the current table, "Combined" looks like the combined "Laminar" and "Turbulent" environments. The table should be corrected for clarity.

Author Response

Thanks for your suggestion! We have removed the last three rows, entitled "Combined", in Table 3 to clarify this question. We label Table 3 in the red color to highlight this change.